Forkhead box D subfamily genes in colorectal cancer: potential biomarkers and therapeutic targets

Chen Ying 1
Qiao Haiyan 2
Zhong Ruiqi 3
Sun Lei 1 sunlei20211013@163.com
http://orcid.org/0009-0008-5174-7581 Shang Bingbing 1 shangbingbing@dmu.edu.cn
1 Emergency Department, The Second Hospital of Dalian Medical University , Dalian , China
2 Laboratory Animal Center, Dalian Medical University , Dalian , China
3 The First Affiliated Hospital of Dalian Medical University , Dalian , China
Altun Zekiye
Electronic publication date: 2024 Oct 29
Publication date: 2024
Volume: 12
Electronic Location ID: e18406
Received 2024 May 10; Accepted 2024 Oct 5
Copyright: © 2024 Chen et al.
Copyright year: 2024
Copyright holder: Chen et al.
License: This is an open access article distributed under the terms of the Creative Commons Attribution License, which permits unrestricted use, distribution, reproduction and adaptation in any medium and for any purpose provided that it is properly attributed. For attribution, the original author(s), title, publication source (PeerJ) and either DOI or URL of the article must be cited.
License URL: https://creativecommons.org/licenses/by/4.0/

Keywords: Forkhead box D subfamily, Colorectal cancer, Immune cells, Prognosis

Funding: Dalian City peak plan This study was supported by Dalian City peak plan. The funders had no role in study design, data collection and analysis, decision to publish, or preparation of the manuscript.

==============================
Background

The forkhead box (FOX) family members regulate gene transcription and expression. FOX family members regulate various biological processes, such as cell proliferation and tumorigenesis. FOXD, a FOX protein subfamily, is associated with poor prognosis for various cancers. However, the potential clinical value of FOXD subfamily members in colorectal cancer (CRC) has not yet been elucidated. Therefore, in this study, we aimed to determine the role of the FOXD subfamily members in CRC development.

Methods

Using HTSeq-count data, clinical data, and single-nucleotide polymorphisms (obtained from The Cancer Genome Atlas Project), and bioinformatics analyses (using DESEQ2 software), we identified differentially expressed genes (DEGs) in CRC. Next, each DEG expression was validated in vitro using reverse transcription-quantitative polymerase chain reaction, western blotting, and immunohistochemistry (IHC).

Results

Among the FOXD subfamily members, the area under the receiver operating characteristic curve of FOXD3 was 0.949, indicating that FOXD3 has a high overall diagnostic accuracy for CRC. Gene Set Enrichment Analysis revealed that FOXD-DEGs were mainly related to pathways such as cytokine, cytokine, and extracellular matrix receptor interactions. Kaplan–Meier curves and nomograms showed that FOXD1, FOXD3, and FOXD4 were prognostically significant. In conclusion, FOXD subfamily members (especially FOXD3) could serve as diagnostic and prognostic biomarkers for CRC and an immunotherapy target in patients with CRC.

Introduction

Colorectal cancer (CRC) is the third most common cancer worldwide, and CRC incidence continues to rise (Sung et al., 2021). In addition, recent epidemiological surveys have revealed that the incidence and mortality rates of CRC have increased in China (Lin et al., 2019). The increase in CRC may be caused by the newly recognized heterogeneous group of cancers termed cancer of unknown primary origin (Rassy et al., 2020). The causes of CRC are complex and diverse, and the risk factors for CRC development include diet, lifestyle, family history, and genetic predisposition (Momenimovahed et al., 2019; Wang et al., 2020). CRC is an age-associated malignancy; in the United States, CRC is most frequently diagnosed in patients aged 65 to 84 (Ioffe & Dotan, 2023). Older patients with CRC are at a higher risk of developing serious postoperative complications; however, there is no consensus on the impact of age on survival outcomes, and prognosis may be influenced by the stage, tumor site, previous comorbidities, and type of treatment (Osseis et al., 2022). Three major molecular pathways are involved in CRC pathogenesis: microsatellite instability acid phosphatase (MSI), chromosomal instability, and the CpG island methylation phenotype (Simons et al., 2013). Immune cell programmed death-ligand 1 (PD-L1) expression is high in MSI-H CRC tumors. Currently, the recommended PD-L1 detection methods include immunohistochemical (IHC) and MSI tests; however, both methods are associated with biological and technical limitations (Adeleke et al., 2022). Despite the development of PD-L1-informed CRC treatment, including novel immunotherapies and targeted therapies, the desired overall survival rates of patients with CRC have not yet been achieved (Lu, Xin & Wang, 2019). The current first-line treatment strategy for treating CRC includes using 5-fluorouracil, leucovorin, and oxaliplatin (FOLFOX) (Johnson et al., 2024). MicroRNAs (miRNAs) play an important role in the development, progression, and metastatic potential of CRC, and act as biomarkers for the prediction of response to anti-vascular endothelial growth factors, anti-epidermal growth factor receptor (EGFR), and FOLFOX in metastatic CRC (Boussios et al., 2019). miRNAs are pivotal regulators of forkhead box (FOX) gene expressions during tumor progression.

The FOX family comprises a set of evolutionarily conserved transcriptional regulatory factors (Shan et al., 2017) that are involved in the regulation of various biological processes, including cell growth, differentiation, proliferation, and apoptosis (Dai et al., 2021; Jin, Liang & Lou, 2020; Laissue, 2019). In mammals, FOX transcription factors are classified into 19 subsets (FOXA-FOXS) based on sequence similarity (Bademci et al., 2019; Zhang et al., 2017). FOXD1-4 are crucial members of the FOX family and promotes cancer cell proliferation in nasopharyngeal carcinoma, non-small cell lung cancer, and breast cancer. In addition, FOXD1 is associated with the pathological differentiation and aggressiveness of CRC (Zong et al., 2022). FOXD 1 may promote the stem cell character and chemotherapy resistance of CRC through direct binding with β-catenin, enhancing of β-catenin nuclear localization (Feng et al., 2023). FOXD2 is located on chromosome 1p32–p34. FOXD2 promotes a new regulatory axis for gastric adenocarcinoma cell proliferation through the Ca2+ signaling pathway mediated by IQGAP3 (Fei, Zhou & Wang, 2023). As a chromatin tuning factor, FOXD2 suppresses CRC growth. FOXD2 methylation is also associated with CRC development (Conesa-Zamora et al., 2015). The current study of FOXD3 and FOXD4 genes focuses on maintaining embryonic stem cell pluripotency, as well as neural crest cell formation, migration, and differentiation, and their interaction with disease pathogenesis (Huang et al., 2024; Wang et al., 2018). FOXD3 is located on human chromosome 1p31 and often acts as a transcriptional suppressor of tumorigenesis in numerous cancer types, including nasopharyngeal carcinoma, lung cancer, and ovarian cancer (Wang et al., 2023). FOXD3 suppresses the proliferation of bone metastatic cell lines in colon cancer via the EGFR/extracellular signal-regulated kinase (ERK) pathway (Wang et al., 2024). In addition, FOXD3 inhibits CRC cell proliferation and migration (Xu et al., 2020). FOXD4 is overexpressed in CRC; FOX4 promotes SW480 cell migration and invasion in vitro (Chen et al., 2018). However, the precise mechanism underlying the role of the FOXD subfamily in CRC remains unclear. Thus, a comprehensive understanding of the mechanism of action of the FOXD subfamily in CRC could offer valuable insights into new therapeutic targets and novel biomarkers for CRC diagnosis. Therefore, in this study, we aimed to determine the role of the FOXD subfamily members in CRC development.

Methods

Data download and pre-processing

Using the TCGAbiolinks package and R software (version 4.0.4) (Colaprico et al., 2016; R Core Team, 2021), we downloaded the following data from The Cancer Genome Atlas (TCGA) database: HTSeq-counts, clinical, and single-nucleotide polymorphisms data for TCGA-Colon Adenocarcinoma Collection (COAD) and TCGA-Rectum Adenocarcinoma (READ). We excluded samples with null values and no survival information, and 643 samples (the baseline data of patients was detailed in Table S1) were included in the study. All the samples in the dataset were derived from Homo sapiens using the Illumina platform. The count data were converted into transcripts per million reads format and log2 conversion was performed. Additionally, we downloaded the gene expression matrix and annotation files of normal tissues from the Genotype-Tissue Expression (GTEx) database. In the bioinformatics analysis, we used strict parameter control. We adopted the Benjamini–Hochberg method to adjust the p-value to control the false discovery rate (FDR) and set adj.p < 0.05 and |log2 fold change| > 1 as the screening threshold.

FOXD subfamily identifications and characteristic analysis

The combined data of TCGA-COAD and TCGA-READ Counts were analyzed by the DESeq2 package (Love, Huber & Anders, 2014) to obtain differentially expressed genes (DEGs). Then, we performed batch survival analysis using the Survival and Survminer packages in R to obtain genes with different expression levels. The FOXD subfamily (FOXD1, FOXD2, FOXD3, and FOXD4) was defined as the primary variable based on the intersection of the above two gene sets. The FOXD subfamily receiver operating characteristic curve (ROC) was analyzed in the pROC package and plotted in the ggplot2 package. In addition, lollipop plots of amino acid changes (gene mutations) in the FOXD subfamily were drawn the using maftools package (Mayakonda et al., 2018).

Functional enrichment analysis of FOXD subfamily

DESeq2 package was used to screen FOXD subfamily-related DEGs with high or low expression. The ggplot2 package was used to draw the volcano maps and the pheatmap package (https://CRAN.R-project.org/package=pheatmap) was used to draw a heat map showing the overall expression pattern of FOXD subfamily-related DEGs. The selection criteria were as follows: p < 0.05 and |log2FC|>1.

Gene Ontology (GO) and Kyoto Encyclopedia of Genes and Genomes (KEGG) enrichment analyses were performed using the clusterProfiler package (Yu et al., 2012), and the significance criterion was set at p < 0.05. For Gene Set Enrichment Analysis (GSEA), we choose “c2. Cp. Kegg. V7.0. Entrez. GMT” as a reference gene set, and the significant concentration threshold values were as follows: FDR < 0.25 and p < 0.05.

Protein–protein interaction analysis and hub gene screening

Protein–protein interaction (PPI) network analysis was performed using the STRING database and Cytoscape software (version 3.8.2). Hub genes were identified using the Molecular Complex Detection (MCODE) plugin (Bader & Hogue, 2003).

Immune cell infiltration analysis

Based on the principle of linear support vector regression, CIBERSORT was used to deconvolve the transcriptome expression matrix and estimate the composition and abundance of 22 immune cells in a mixed cell sample, as previously described (Newman et al., 2015). Samples with p < 0.05 were filtered to generate an immune cell infiltration matrix. Correlation heat maps were drawn using R-package corrplot (Yang et al., 2021) to visualize the interrelationship between the 22 immune cells, and lollipop maps were drawn using the GSVA package (Hänzelmann, Castelo & Guinney, 2013) to show the correlation of 22 immune cells with the FOXD gene family.

Survival analysis and clinical relevance

Prognostic survival analysis was performed using the Survminer and Survival packages. To determine the thresholds for the high and low expression groups, raw gene expression data were collected and normalized using log2 transformation to reduce data discretization and median centralization. Next, we determined the tangency point of the greatest difference by calculating the correlation between patient survival data and gene expression levels using the ‘surv_cutpoint’ function (R package ‘survminer’), which divides the expression data based on minimum-tangency maximization statistics. To ensure the reliability of results, the robustness of the selected tangent points was verified using various methods such as resampling and cross-validation. The Kaplan–Meier method was used to draw the survival curve, and the log-rank test was used to compare the survival differences between the high and low expression groups. In addition, multivariate Cox proportional hazards model analyses included age, sex, and other clinical characteristics as covariates to adjust for confounders (The detailed parameters are shown in Table S2). To plot a concordance curve, we fitted the nomogram and performed calibration analysis.

Immunohistochemistry analysis

CRC tumor tissues and adjacent normal colon tissues (n = 28) were collected from the Second Hospital of Dalian Medical University. All participants were informed of the reasons to conduct the study, anonymity was ensured, and how the collected data were stored. We received written informed consent from participants of our study. Eight cases with missing information and unqualified section quality were excluded, and 20 samples were retained. The protocol was approved by the Second Hospital of Dalian Medical University ethics committee (approval number: 2021-079). FOXD subfamily gene expression in the collected samples was measured using IHC. Human CRC and adjacent normal colon tissues were fixed overnight in 10% formalin and embedded in paraffin. The tissue blocks were cut into 3 μm tissue sections. The tissue sections were dewaxed, rehydrated, and subjected to antigen retrieval in either citrate or EDTA buffer, depending on the antibody. Following 3% hydrogen peroxide treatment, the sections were incubated overnight at 4 °C with the following primary antibodies: anti-FOXD1 (1:100 dilution, ab129324; Abcam, Cambridge, UK), anti-FOXD2 (1:50; TA322739, OriGene, Rockville, Maryland, US), anti-FOXD3 (1:100 dilution, abs133773; Absin, Shanghai, China), and anti-FOXD4 (1:100, orb156929; Biorbyt, Cambridge, UK). The bound antibodies were detected using horseradish peroxide-conjugated secondary antibodies and developed with nickel-diaminobenzidine using a two-step test kit (PV-9000; Beijing Zhongshan Jinqiao Biological, Beijing, China), according to the manufacturer’s instructions. IHC scores were calculated through multiplying the staining score by the percentage score. The percentage of positive cells were judged as 0, ≤5% positive; 1, 6% to 25% positive; 2, 26% to 50% positive; 3, 51% to 75% positive; and 4, 76% to 100% positive. Staining intensity was judged as 0 for no staining, 1 for light yellow or yellow, 2 for brown, and 3 for dark brown (Gao et al., 2020; Khan et al., 2018; Mo et al., 2021). The IHC staining results were assessed by one senior pathologist. The final staining score ranged from 0 to 12, less than six was considered as low expression, while staining score of six or more was considered as high expression.

Cell culture

CCD841CON cells were obtained from Meisen CTCC (Zhejiang, China) and cultured in 90% Dulbecco’s Modified Eagle Medium (DMEM) supplemented with 1% penicillin/streptomycin (PS) and 10% fetal bovine serum (FBS). All colon cancer cell lines (SW480, SW620, HT29 and HCT 116) were purchased from ProCell company (Wuhan, China). Among of them, SW480 and SW620 cells were cultured in 90% DMEM supplemented with 10% FBS and 1% PS. HT29 and HCT116 cells were cultured in 90% McCoy’s 5A medium supplemented with 10% FBS with 1% PS. The cells were incubated at 37 °C in a humidified 5% CO2 incubator.

Western blot assay

Total proteins were extracted from cells using radioimmunoprecipitation assay. The concentrations of the protein samples were determined using a bicinchoninic acid protein concentration assay. The prepared 10% sodium dodecyl sulfate-polyacrylamide gel was poured onto a glass plate and allowed to solidify. Next, 50 µg of the marker and each protein sample was loaded into the gel wells. Subsequently, the samples were electrophoresed under reducing conditions and transferred onto polyvinylidene fluoride membranes. The membranes were blocked for 2 h. Next, the samples were incubated with primary antibodies used were against FOXD1 (dilution 1:1000, abs102294; Absin, Shanghai, China), FOXD2 (dilution 1:1000, TA322739; Origene, Rockville, MD, US), FOXD3 (dilution 1:1000, abs133773; Absin, Shanghai, China), and FOXD4 (dilution 1:1000, orb156929; Biorbyt, Cambridge, UK) overnight at 4 °C. Following incubation, the membrane was washed thrice with Tris-buffered saline (TBS) 0.1% Tween 20 (TBST) for 5 min each. Next, the membranes were placed in a diluted secondary antibody solution and incubated at room temperature for 2 h. After incubation with the secondary antibody, the membrane was washed with TBST three times for 10 min each and then washed once with TBS for 10 min. All the bands were detected using the ECL detection kit (SW 134; Seven), according to the manufacturer’s protocol. The results were analyzed using the ImageJ software. All experiments were performed in triplicate.

RNA isolation and quantitative real-time polymerase chain reaction (qRT-PCR)

We used TransZol Up (ET111-01-V2; TransGen Biotech, Beijing, China) to extract total RNA from the cell lines (normal colon epithelial cells and four colorectal cancer cell lines), five freshly resected human colon cancer tissues, and seven adjacent normal colon tissues. The A260/A280 ratio of the extracted RNA ranged from 1.8–2.0. We used Cytation three image reader to detect the total RNA concentration and added an appropriate amount of RNase-free water to obtain a concentration of 1 ug/ul. The experiment was performed according to the manufacturer’s instructions (Chen et al., 2022). The primer sequences used are listed in Table S6. cDNA was synthesized from the extracted RNA samples (1 µl) using a PerfectStart ® Uni RT&qPCR kit (AUQ-01-V2; TransGen Biotech, Beijing, China), according to the manufacturer’s instructions and qPCR was performed using a Applied Biosystems StepOne Real-Time PCR kit (AQ311-01; TransGen Biotech, Beijing, China). We used qPCR to detect differential expression of the FOXD subfamily in cells and tissues. The relative gene expression of the FOXD subfamily was calculated using the 2−ΔΔCT method. Expression was normalized to that of GADPH. The qPCR was performed in triplicate.

Statistical analysis

All data analyses were performed using R (version 4.0.4, http://r-project.org/). To ensure reliability of results, t-test, chi-square test, Kaplan–Meier survival analysis, and Cox proportional risk model were performed. The t-test was used to compare continuous variable (such as gene expression values) differences between the two groups, and the chi-square test was used to analyze the associations between categorical variables, such as the relationship between gene mutation status and clinical characteristics. Kaplan-Meier survival analysis was used to assess the difference in survival between different groups (e.g., high and low gene expression groups), and significance was determined by log-rank test. The Cox proportional risk model was used in multivariate survival analysis to assess the combined impact of multiple variables (gene expression level and clinical features) on survival outcomes. Statistical analyses of qPCR and western blotting were performed using GraphPad Prism software (version 5.0). The significance threshold for all statistical analyses was set at p < 0.05, unless otherwise specified.

Results

Characteristics of the FOXD subfamily

We determined the diagnostic potential, and mutational characteristics of FOXD1, FOXD2, FOXD3, and FOXD4 in CRC. The area under the curve (AUC) of the receiver operating characteristic (ROC) was calculated. The AUC values for FOXD1, FOXD2, FOXD3, and FOXD4 were 0.732, 0.784, 0.949, and 0.750, respectively (Figs. 1A–1D). With an AUC value of 0.949, FOXD3 was the most significant diagnostic biomarker for CRC. Compared to known CRC markers such as CEA and CA19-9, FOXD3 demonstrated greater diagnostic accuracy.

Figure 1 The characteristics of FOXD subfamily genes.

(A–D) ROC-curve analysis of FOXD1, FOXD2, FOXD3, and FOXD4, respectively. The AUC of FOXD1, FOXD2, FOXD3, and FOXD4 was 0.732, 0.784, 0.949, and 0.750, respectively. (E–H). The mutation analysis of the FOXD subfamily is shown in lollipop plots. The mutation of FOXD1 was not detected. FOXD2 was mutated in both COAD and READ. FOXD3 and FOXD4 were mutated only in COAD. Green dots represent missense mutations and blue dots represent gene deletions.

Mutation analysis of the FOXD subfamily were visualized in lollipop plots (Figs. 1E–1H). FOXD2 was mutated in both COAD and READ, FOXD3 and FOXD4 were mutated only in COAD, and FOXD1 mutations were not detected. Green dots represent missense mutations and blue dots represent gene deletions. These results provide biological clues regarding the characteristics and functions of FOXD in CRC.

Identification and visualization of FOXD subfamily-related DEGs

We divided the FOXD subfamilies into two groups based on the median DEG expression values. These two groups were the high- and low-expression groups. We performed differential gene expression analysis to identify the FOXD subfamily related DEGs. The results are represented in volcano plots (Figs. 2A–2D). We identified 590, 1,388, 737, and 1,122 FOXD1-4-related DEGs, respectively. Up- and down-regulated genes are represented in red and blue, respectively. The top 50 up- and down-regulated genes are depicted in heat maps (Fig. 2E–2H). Blue represents low FOXD expression and red represents high FOXD expression.

Figure 2 Visualization of FOXD subfamily genes difference analysis.

(A–D) Volcano plots showed differentially expressed genes comparing high vs. low expression in the FOXD subfamily. Specifically, (A) represents FOXD1, (B) represents FOXD2, (C) represents FOXD3, and (D) represents FOXD4, respectively. Down-regulated genes were in blue and up-regulated genes were in red. (E–H) Heat maps showed that FOXD subfamily genes were grouped into low and high expression. Blue indicates low gene expression and red indicates high gene expression.

Functional enrichment analysis of FOXD subfamily-related DEGs

Next, we performed GO/KEGG analysis of the FOXD subfamily related DEGs (Fig. 3). GO analysis (Table S3) revealed correlations between FOXD1-related DEGs and metabolic development. FOXD2-related DEGs were associated with humoral antimicrobial responses and cornification. FOXD3-related DEGs were associated with axonogenesis and the modulation of chemical synaptic transmission. FOXD4-related DEGs were associated with the formation of quadruple SL/U4/U5/U6 snRNPs and mRNA transsplicing. KEGG pathway analysis (Table S4) revealed that FOXD2-related DEGs were linked to cytokine receptor pathways. Moreover, FOXD3-related DEGs were associated with the calcium signaling pathway. FOXD4-related DEGs correlated with salivary secretion and GABAergic synaptic pathways. No pathway enrichment was observed for FOXD1-related DEGs.

Figure 3 Functional enrichment analysis of the differentially expressed genes (DEGs) of the FOXD subfamily.

(A–D) The network graph showed the GO enrichment analysis of FOXD1, FOXD2, FOXD3, and FOXD4-related DEGs. (E–G) Bubble plots showed the KEGG pathways of FOXD subfamily-related DEGs.

Gene set enrichment analysis

We subsequently performed GSEA of the FOXD subfamily related genes (Fig. 4, Table S5). GSEA showed that the FOXD subfamily-related genes were mainly linked to KEGG pathways, including cytokine-cytokine receptor interactions and ECM-receptor interactions.

Figure 4 GSEA of FOXD subfamily-related genes.

(A–F). The results showed that cytokine-cytokine receptor interaction, ECM-receptor interaction, focal adhesion, natural killer cell-mediated cytotoxicity, oxidative phosphorylation, and ribosome were the main KEGG pathways of FOXD subfamily-related genes, respectively.

Protein-protein interaction analysis and Hub genes identification

The FOXD subfamily-related DEGs were uploaded to the STRING database for PPI network analysis. The results were then imported into Cytoscape. DEGs were shown in different colors based on logFC expression values (Figs. 5A–5D, top panel). Red and blue represented the up- and down-regulated genes, respectively. The deeper the color, the larger the |logFC| value. Among the FOXD subfamily, FOXD2 and FOXD3 had more hub genes. Fig. 5A showed that the hub genes of FOXD1 were KRT5, KRT6A, KRT6B, KRT6C, KRT7, KRT14, KRT16, and KRT24. There were 21 hub genes of FOXD2, all of which were down-regulated (Fig. 5B). FOXD3 had 13 hub genes, all of which were also down-regulated, such as CXCL5, NYP, ADCY2, and so on (Fig. 5C). SPRR1A, SPRR1B, SPRR2E, SPRR2F, and SPRR3 were the hub genes of FOXD4 (Fig. 5D).

Figure 5 The PPI network of hub genes.

(A) PPI visualization of FOXD1-DEGs and hub genes was displayed. (B) PPI visualization of FOXD2-DEGs and hub genes was displayed. (C) PPI visualization of FOXD3-DEGs and hub genes was displayed. (D) PPI visualization of FOXD4-DEGs and hub genes was displayed.

Analysis of immune cell infiltration

We further estimated the changes in immune cell infiltration in CRC using the CIBERSORT algorithm (Fig. 6A). The relationship between the levels of immune cell infiltration is shown in red and blue, with blue indicating a positive correlation and red indicating a negative correlation (Fig. 6B). The correlations between immune genes and FOXD1, FOXD2, FOXD3 and FOXD4 are shown as lollipop plots. The deeper the color, the smaller the p-values, and the stronger the correlations (Fig. 6C–6F).

Figure 6 Immune cell infiltration analysis.

(A) The immune landscape represented the overall immune infiltration of patients. (B) The correlation heatmap of the 22 immune cells. Blue indicated positive correlation, red indicated negative correlation, and the darker the color, the stronger the correlation. (C–F) Lollipop figures showed the correlation between immune cells and FOXD subfamily genes.

Survival analysis and clinical correlation analysis of FOXD subfamily genes

We performed a prognostic survival analysis according to the high- and low-expression groups of FOXD subfamily genes (Fig. 7A–7D). The relationship between the levels of immune cell infiltration is shown, and positive and negative correlations are represented as blue and red, respectively. The clinical correlation parameters included tumor T-stage, N-stage, M-stage, sex, age, pathologic stage, history of colonic polyps, lymph node invasion, and FOXD subfamily genes and are displayed in a nomogram (Figs. 7E–7H). Using these nomograms, we predicted 1-year, 3-year, and 5-year patient survival prognoses. Finally, the clinical prediction models were assessed using calibration curves (Figs. 7I–7L). The closer the curves of 1-year, 3-year, and 5-year to the grey ideal line, the better the predictions.

Figure 7 Survival analysis and clinical correlation analysis of FOXD subfamily genes.

(A–D) The prognostic survival analysis of FOXD1, FOXD2, FOXD3, and FOXD4. (E–H) The clinical correlation analysis of FOXD subfamily genes combined with clinical indices. (I–L). The 1-, 3-, and 5-year nomogram calibration curves of FOXD1, FOXD2, FOXD3, and FOXD4 were displayed, respectively.

Analysis of FOXD subfamily expression by IHC staining of patient samples

In human colon cancer tissues, protein expression levels of the FOXD subfamily were determined using IHC. FOXD1 expression was not detected in both tumor and matched normal tissues. The moderate staining was observed in FOXD2 and FOXD4. FOXD3 was strong staining. Overall, FOXD2, FOXD3 and FOXD4 were expressed at higher levels in tumors, compared with the adjacent normal tissue (Fig. 8).

Figure 8 The IHC analysis of FOXD subfamily genes.

(A) FOXD1 expression was not detected in both tumor and normal tissues. (B) FOXD2 showed intermediate intensity staining in tumor tissues. (C) FOXD3 showed strong staining in tumor tissues. (D) FOXD4 showed intermediate intensity staining in tumor tissues. Images were at 20× magnification, and insets were at 40× magnification. CRC tumor tissues and adjacent normal colon tissues were collected from the Second Hospital of Dalian Medical University.

Identification of FOXD subfamily gene expression in vitro

Western blotting results showed that FOXD1 expression was higher in all CRC cell lines than normal colon cells (CCD841CON). Among them, the difference between SW620 cell line and CCD841CON cell line was the most significant (Fig. 9A). Compared to the CCD841CON cell line, FOXD2 was expressed at a higher level in SW480, SW620, HT29, and HCT116 cell lines. However, the difference was statistically significant only in the SW480, SW620, and HCT116 cell lines (Fig. 9B). FOXD3 expression seemed to be higher in all CRC cell lines, but the difference was statistically significant only in the HT29 cell line (Fig. 9C). In addition, the FOXD4 expression level seemed to be higher in all CRC cell lines, but the difference was statistically significant only in the SW620 cell line (Fig. 9D). Next, we examined the expression of FOXD1 and FOXD4 in tissues and cell lines using qPCR (Fig. 10). The results showed that the expression levels of FOXD1 and FOXD4 were higher in tumors compared with the paracancerous tissues (Figs. 10A, 10C). However, FOXD1 expression seemed to be higher in SW480, HT29, and HCT116 cell lines than in normal colon cells (CCD841CON) but the difference was not statistically significant (Fig. 10B). The expression level of FOXD4 was higher in all CRC cell lines than in normal colon cells, but the difference was statistically significant only in the SW620 and HT29 cell lines (Fig. 10D). The list of primer sequences was shown in Table S6.

Figure 9 Cell lines were assessed for FOXD1, FOXD2, FOXD3, and FOXD4 expression by WB.

(A) FOXD1 expression was higher in all CRC cell lines, and there was a statistically significant difference. (B) FOXD2 was expressed at a higher level in SW480, SW620, HT29, and HCT116 cell lines than in normal colon cells (CCD841CON), but the difference was statistically significant only in the SW480, SW620, and HCT116 cell lines. (C) FOXD3 expression seemed to be higher in all CRC cell lines, but the difference was statistically significant only in the HT29 cell line. (D) The FOXD4 expression level seemed to be higher in all CRC cell lines, but the difference was statistically significant only in the SW620 cell line. CCD841CON was purchased from Meisen CTCC (Zhejiang, China), and all colon cancer cell lines were purchased from ProCell company (Wuhan, China). An asterisk (*) indicates a P-value < 0.05, suggesting the result is statistically significant. Two asterisks (**) indicate a P-value < 0.01, demonstrating a high level of statistical significance.

Figure 10 Expression of FOXD1 and FOXD4 was assayed with qPCR.

(A) The expression level of FOXD1 was higher in tumors compared with the paracancerous tissues. (B) FOXD1 expression was higher in SW480, HT29, and HCT116 cell lines than in normal colon cells (CCD841CON), but the difference was not statistically significant. (C) FOXD4 was expressed at a higher level in tumor tissues than in paracancerous tissues. (D) FOXD4 was expressed at a higher gene level in CRC cells (HCT116, HT29, SW620, SW480) than in normal colon cells (CCD841CON). However, the difference was statistically significant only in the SW620 and HT29 cell lines. An asterisk (*) indicates a P-value < 0.05, suggesting the result is statistically significant. Two asterisks (**) indicate a P-value < 0.01, demonstrating a high level of statistical significance.

Discussion

The FOX protein is a transcription factor that contains a wing helix DNA-binding domain (DBD). All members of the FOX family share this DBD but have different deactivation and inhibition domains (Bach et al., 2018). Specifically, the FOX family participates in cancer maintenance, progression, and metastasis at various levels. There were at least 14 FOX subgroups have been reported to take part in the pathogenesis of CRC. Among of them, FOXO subfamily, FOXM1 and FOXP3 were regarded as important regulators of CRC tumorigenesis (Laissue, 2019). In this study, we analyzed the expression characteristics and clinical significance of FOXD family genes (FOXD1, FOXD2, FOXD3, and FOXD4) in CRC using bioinformatics analyses (Chen et al., 2023). FOXD1, also known as FKHL8, FREAC4, and FREAC-4, is a protein encoding gene. It plays an important role in the regulation of cell reprogramming and influences the development of cancer cells in various cancers (Liu et al., 2024). In addition, it is associated with cell proliferation, epithelial-mesenchymal transformation, and poor clinicopathological outcomes (Prasad et al., 2024; Sun et al., 2023). Knockdown of FOXD1 could significantly inhibite the colony-forming ability of oral cancer cells after radiotherapy and enhance the expression of TXNIP via the JAK-STAT signalling pathway (Lin et al., 2020). In basal-like breast cancer, FOXD1 could regulate enhancer gene programs associated with tumor progression and inhibition of FOXD1 could reduce metastasis by inactivating EMT-associated enhancers (Kumegawa et al., 2023). FOXD1 acts as a carcinogenic transcription factor in cancers such as metastatic melanoma and renal clear cell sarcoma (Bond et al., 2021; Li et al., 2019; Pan, Li & Chen, 2018; Zhang et al., 2020; Zhao et al., 2015). The overall survival rate of melanoma cells was reduced by high FOXD1 expression, which was positively associated with resistance to BRAFi or a combination of BRAFi and MEKi. FOXD1 knockdown restores sensitivity to BRAFi resistance (Sun et al., 2021). Similarly, FOXD1 and Gal-3 interact to form a positive feedback loop, which is a potential therapeutic target in lung cancer (Li et al., 2019). Moreover, FOXD1 activates the ERK1/2 signaling pathway and promotes CRC development (Pan, Li & Chen, 2018).

FOXD2, which is also known as FKHL17 or FREAC9, contains a distinct forkhead domain. It acts as a chromatin regulator that performs various functions in different chromatin environments, enabling cells to respond effectively to environmental signals. In CRC, FOXD 2 plays a role in the fine-tuning of gene expression programs to inhibit tumor growth (Kim et al., 2023). A previous study showed that FOXD2 induced the expression of cAMP-dependent protein kinase RI α through its synergistic effect with protein kinase B, thereby increasing cAMP sensitivity (Johansson et al., 2003). Okabe et al. (2019) found that the expression level of FOXD2 mRNA is high in normal podocytes and that FOXD2 maintains podocyte integrity. Notably, only three human studies on FOXD2 have been linked to cancer. Both meningioma and CRC are associated with FOXD2 (Conesa-Zamora et al., 2015).

FOXD3 is an acidic, unstable, and hydrophilic protein that does not belong to the membrane or secretory protein. It has the greatest possibility of being localized in the nucleus, affecting the regulation of many biological processes (Lam et al., 2013). Abnormal FOXD3 expression has been observed in melanomas, neuroblastomas, and other cancers. Thus, FOXD3 may exert antitumor effects. Additionally, the Ras/Raf/MEK/ERK pathway is activated by FOXD3 silencing (Li et al., 2017). MAPK (ERK), c-Jun N-terminal kinase (JNK), and p38 kinase play regulatory roles in tumors. Constitutive MAPK activation triggers chemotherapy resistance in the cancer cells of multiple human malignancies. The RAS/RAF/MEK/ERK pathway, a regulator of various cellular activities such as cell proliferation and death, is a key driver in human cancers (Gao et al., 2019; Nichols et al., 2018; Sanchez et al., 2019). The inhibition of the MAPK pathway may be a potential research direction for early treatment. Furthermore, promoting the expression of FOXD3 is beneficial for the prognosis of patients with neuroblastoma (Li et al., 2013), whereas inhibiting FOXD3 expression is detrimental to the prognosis of breast cancer patients (Chu et al., 2014). FOXD3 is hypermethylated in CRC which indicated FOXD3 could act as a diagnostic biomarker (Hauptman et al., 2019).

FOXD4, also known as FKHL9, FOXD4A, or FREAC5, is a transcription factor. As protein-coding genes, chromosome 9P deletion syndrome and cerebral palsy (Chen et al., 2012; Humphray et al., 2004) correlate with FOXD4. Interestingly, FOXD4 is associated with phenotypes such as obsessive-compulsive disorder and suicidal ideation (Minoretti et al., 2007) and is not conducive to CRC prognosis (Chen et al., 2018; Li, Li & Liao, 2020). Additionally, FOXD4 methylation is associated with immune dysfunction and cell proliferation (Liu et al., 2021). Thus, FOXD4 may act as an independent prognostic marker for COAD (Xie, Wang & Peng, 2023).

In this study, although we also used bioinformatics methods to identify FOXD subfamily genes, our approach was different. Through GO and KEGG enrichment analyses, we found that FOXD family related DEGs were significantly enriched in a variety of biological processes and signaling pathways. FOXD1-DEGs are associated with keratinization and development, FOXD2-DEGs with antimicrobial humoral reactions, FOXD3-DEGs with axonogenesis and synaptic transmission regulation, and FOXD4-DEGs with mRNA transcription splicing. These results suggest that the FOXD family genes may influence the occurrence and development of CRC by regulating various biological processes and signaling pathways. GSEA showed that FOXD family genes were mainly related to cytokine-cytogenic receptor interactions, extracellular matrix-cell adhesion, natural killer cell-mediated cytotoxicity, oxidative phosphorylation, and ribosome pathways. These pathways are widely involved in remodeling of the tumor microenvironment, metabolic reprogramming of tumor cells, and immune escape mechanisms, suggesting that the FOXD gene family may play an important role in the regulation of CRC. We screened hub genes related to the FOXD family genes using PPI analysis. These Hub genes may play key roles in CRC occurrence and development. For example, FOXD4 was enriched in pathways related to signal transduction and transcriptional regulation, suggesting that it may play an important role in tumor signaling and gene expression regulation. Immune cell infiltration analysis revealed that FOXD family genes were significantly associated with the infiltration of multiple immune cells. In particular, FOXD3 and FOXD4 were associated with the infiltration of effector T and natural killer cells, suggesting that the FOXD family of genes may affect the immune escape and therapeutic response of CRC cells by regulating the immune microenvironment. In future studies, the specific mechanism of the FOXD gene family in CRC immune regulation should be further explored to identify new targets for improving the efficacy of immunotherapy. Kaplan-Meier curves were used to analyze the overall survival of patients, we found that high FOXD1, FOXD3 and FOXD4 expression was associated with a poor prognosis, which is consistent with existing studies on several genes with similar functions. For example, studies have shown that high FOXM1 expression is associated with poor prognosis in a variety of cancers (Tabnak et al., 2023), suggesting that FOXD family genes may follow a similar mechanism in CRC. Subsequently, a nomogram prognostic model of the FOXD subfamily for patients with CRC was developed by combining clinical indicators. This model can be used to predict the survival of patients at one, three, and 5 years, thereby providing a basis for clinical decision-making. The calibration curve revealed that the 3-year nomogram calibration curve had good predictive power, suggesting FOXD family genes may serve as new therapeutic targets. By targeting and regulating the expression and function of these genes, we expect to develop new therapeutic strategies for CRC with potential value in personalized medicine and targeted therapies.

Additionally, we identified several confounding factors that may have affected the results of this study. To control and correct for these factors, we developed various strategies to ensure the reliability and accuracy of the results. First, considering the heterogeneity of the sample processing and sequencing methods, we standardized the samples for data preprocessing and used DESeq2 to correct the batch effect. Second, for differences in patient age, sex, and ethnicity, we used multiple regression models to adjust for the analysis and introduced these clinical variables as covariates in the functional enrichment and survival analyses. In addition, we controlled for heterogeneity across tumor subtypes and stages through stratified analyses and used strict statistical thresholds and multiple test correction methods, such as the Benjamini-Hochberg method, to control for false-positive rates. Finally, the levels of immune cell infiltration were assessed using the CIBERSORT algorithm and considered in the correlation analyses to reduce the impact on the tumor microenvironment. Despite these strategies, it is necessary to further validate the findings in independent samples and different clinical settings to ensure their universality and clinical application potential.

Although this study is based on an in-depth analysis of data from TCGA and GTEx databases, there are still some limitations. First, there may be systematic errors in sample processing and sequencing methods at different research centers, and certain key clinical features (such as treatment and recurrence) may be inadequately documented, which may have affected the interpretation and application of the results. Second, the uneven distribution of race, sex, and age in the sample population may limit the generalizability of the results across different populations. In addition, despite the use of multiple bioinformatics tools and algorithms, the results still depend on input data and parameter settings, which may affect their repeatability and reliability. Third, the CIBERSORT tool has some limitations: it assumes that the gene expression profile in all samples is relatively stable and consistent, but the actual sample may have high heterogeneity, thus affecting the analysis results; CIBERSORT relies on standard reference gene expression matrices that may differ from actual samples, resulting in limited accuracy of the results (this difference may be even more pronounced in different tumor types); CIBERSORT applies reference gene expression matrices, possibly affecting accuracy of the results if the gene expression profiles of immune cells in an actual sample do not match these matrices; and in the qPCR experiment, difference is expression profiles may have been due to the primers used. Although the primer sequences for FOXD2 and FOXD3 were redesigned, we were unable to determine the PCR Ct values. Therefore, it is necessary to verify this in an independent cohort in the future, further confirm the specific mechanism of action of FOXD family genes in CRC through functional experiments in cell and animal models, and verify their prognostic value and therapeutic potential through multicenter clinical studies.

Conclusions

In conclusion, FOXD subfamily members serve as strong potential biomarker for CRC prognosis prediction. In addition, FOXD subfamily members could serve as novel therapeutic targets.

Supplemental Information

Supplemental Information 1 Raw data.

Supplemental Information 2 qPCR raw data.

Supplemental Information 3 The IHC score of FOXD subfamily genes.

Supplemental Information 4 Baseline data of patients.

Supplemental Information 5 Univariate and multivariate Cox regression analysis of FOXD subfamily.

Supplemental Information 6 The results of GO enrichment analysis.

Supplemental Information 7 KEGG pathway enrichment analysis.

Supplemental Information 8 The results of Gene Set Enrichment Analysis (GSEA).

Supplemental Information 9 A list of primer sequences.

Supplemental Information 10 MIQE checklist.

The authors are deeply grateful to the patients for their cooperation. We also thank HELIX for article revision and language editing.

Additional Information and Declarations

Competing Interests

Author Contributions

Human Ethics

Data Availability

The authors declare that they have no competing interests.

Ying Chen performed the experiments, analyzed the data, prepared figures and/or tables, authored or reviewed drafts of the article, and approved the final draft.

Haiyan Qiao performed the experiments, analyzed the data, authored or reviewed drafts of the article, and approved the final draft.

Ruiqi Zhong analyzed the data, authored or reviewed drafts of the article, and approved the final draft.

Lei Sun conceived and designed the experiments, prepared figures and/or tables, and approved the final draft.

Bingbing Shang conceived and designed the experiments, prepared figures and/or tables, authored or reviewed drafts of the article, and approved the final draft.

The following information was supplied relating to ethical approvals (i.e., approving body and any reference numbers):

The Second Hospital of Dalian Medical University granted Ethical approval to carry out the study within its facilities (Ethical Application Ref: 2021-079)

The following information was supplied regarding data availability:

The baseline data of patients was detailed in Table S1. Univariate and multivariate Cox regression analysis of FOXD subfamily was detailed in Table S2. The results of GO enrichment analysis and KEGG pathway enrichment analysis were detailed in Tables S3 and S4, respectively. Gene Set Enrichment Analysis was detailed in Table S5. A list of primer sequences of FOXD subfamily genes was detailed in Table S6. The IHC score was detailed in the Supplemental Information.

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
