# Peer review of "Forkhead box D subfamily genes in colorectal cancer: potential biomarkers and therapeutic targets"

_PeerJ, doi:10.7717/peerj.18406_

## Round 0.1 · original submission · Minor Revisions

In this article, the authors used multiple bioinformatics methods and obtained data from TCGA to study the relationship between the FOXD subfamily and CRC. The article is well written in a comprehensible manner. However, there are sections that need to be addressed and edited.

·

Basic reporting

1. The manuscript require a thorough grammatical check and language correction

2. Figures labeling should be more detailed to provide more specified information given in the figures.
Example: in figure 2 author have written “(A)-(D). Volcano plot showing differentially expressed genes comparing high vs. low expression in FOXD subfamily. Down-regulated genes are in blue and up-regulated genes are in red.”
Where it should be specified that “(A)-(D) represents FOXD1- FOXD4 respectively.

Experimental design

1. In the section 2.1 of the manuscript, authors should provide the inclusion and exclusion criteria for the dataset.

2. Line 125 states that 28 samples were collected however in discussion section line 312 it is given that 20 samples were used. Kindly justify.

3. Line 150 states “5 adjacent cancerous tissues and 7 cancerous tissues”. Explain.

4. What was taken as control for IHC and how the expression was analyzed to be considered as high or low?

5. The Real Time PCR is performed for FOXD1 and FOXD4 but not for FOXD2 and FOXD3. Why?

Validity of the findings

1. Authors had given a lengthy background of FOXD members in discussion section however discussion lacks the correlation of the findings with previous researches. Author should adopt a more comparative approach to provide better insight to his findings with elaborate references.

Additional comments

Overall the study appears to be technically sound and authors have provided detailed analysis of FOXD subfamily genes in colorectal cancer. However, addressing the mentioned comments is necessary for authors.

Reviewer 2 ·

Basic reporting

Title: The Clinical Value of FOXD Subfamily Genes in Colorectal Cancer

Abstract:
The abstract provides a concise summary of the study, including the objective, methods, results, and conclusion. It effectively outlines the importance of the FOXD subfamily genes in colorectal cancer and the methodologies used to analyze their clinical significance.

Introduction:

Background: The introduction gives a general overview of colorectal cancer (CRC) and the importance of identifying reliable biomarkers for diagnosis, prognosis, and treatment. The focus on the FOXD subfamily genes is justified, though the background information could be expanded to provide a deeper context.
Literature Review: The current state of research on the FOXD subfamily genes in CRC is briefly mentioned. More detailed discussion and citations of recent studies would strengthen the introduction.
Materials and Methods:

Data Sources: The study uses datasets from TCGA, and various bioinformatics tools are employed for data analysis. The selection criteria and characteristics of these datasets are not fully detailed.
Analytical Methods: The manuscript describes the use of DESEQ2 for differential expression analysis, survival analysis, ROC curves, and functional enrichment analysis. Experimental techniques include RT-qPCR, western blotting, and immunohistochemistry.
Results:

Gene Expression and Prognosis: The results section presents findings on the differential expression of FOXD subfamily genes in CRC, their mutation profiles, and their association with survival outcomes.
Diagnostic and Prognostic Value: The diagnostic accuracy of FOXD3 is highlighted using ROC curves.
Immune Infiltration: The association between FOXD gene expression and immune cell infiltration is analyzed using CIBERSORT.
Figures and Tables:

The figures and tables are relevant and effectively illustrate the data. However, some figures could be of higher quality and more clearly labeled.
Supplementary data and raw data files are mentioned but should be explicitly referenced and discussed in the text.
Discussion:

Interpretation of Results: The discussion interprets the findings and their implications for CRC diagnosis and treatment.
Comparison with Previous Studies: There is a need for a more thorough comparison with existing literature on the FOXD subfamily in CRC.
Limitations: The study's limitations are not extensively discussed, which is necessary for a balanced interpretation of the results.
Conclusion:
The conclusion summarizes the clinical significance of the FOXD subfamily genes in CRC and suggests potential future research directions.
Language and Style:

The manuscript is generally well-written but would benefit from thorough proofreading to correct grammatical errors and improve clarity.
Some phrases and sentences could be restructured for better readability.
Ethical Considerations:

The manuscript lacks a clear statement on ethical approval and consent for the use of human tissue samples. This is critical for ensuring ethical compliance.
References:

The references are relevant and up-to-date, but additional citations of recent studies on the FOXD subfamily genes in CRC would strengthen the manuscript.
Summary
The manuscript is well-structured and addresses an important topic in colorectal cancer research. While the study is comprehensive and methodologically sound, it would benefit from a more detailed background, improved language clarity, explicit ethical considerations, and a thorough discussion of its limitations. Incorporating these elements would enhance the overall quality and impact of the manuscript.

Experimental design

Objective:
The objective of the study is to investigate the clinical value of FOXD subfamily genes in colorectal cancer (CRC) by analyzing their expression, mutation profiles, diagnostic and prognostic significance, and association with immune cell infiltration.

Data Sources:
Datasets: The study utilizes datasets from The Cancer Genome Atlas (TCGA) for CRC.
Selection Criteria: The criteria for selecting datasets and samples are not explicitly detailed, which could impact the reproducibility and validity of the results.
Bioinformatic Analysis:
Gene Expression Analysis:
Tool: DESEQ2 is used for differential expression analysis.
Detailing: While DESEQ2 is a robust tool, the manuscript should explain why this tool was chosen over others and provide more detail on the parameters and thresholds used for analysis.
Survival Analysis:
Method: Kaplan-Meier method and Cox proportional hazards model are employed to assess the association between gene expression and patient survival.
Detailing: The manuscript should provide details on how the cutoff values for high and low expression groups were determined.
Diagnostic Accuracy:
Tool: Receiver Operating Characteristic (ROC) curves are used to evaluate the diagnostic potential of FOXD subfamily genes.
Detailing: The manuscript highlights the AUC value for FOXD3, but more detailed interpretation and comparison with existing biomarkers are needed.
Immune Infiltration Analysis:
Tool: CIBERSORT is used to analyze the relationship between FOXD gene expression and immune cell infiltration.
Detailing: The manuscript should describe the preprocessing steps for the CIBERSORT analysis and discuss the limitations of using this tool.
Experimental Validation:
RT-qPCR:
Purpose: Used to validate the expression levels of FOXD subfamily genes in CRC tissue samples.
Detailing: The manuscript should provide details on the primers used, the number of replicates, and the normalization method.
Western Blotting:
Purpose: Used to validate the protein expression of FOXD subfamily genes.
Detailing: Information on the antibodies used, loading controls, and quantification methods should be included.
Immunohistochemistry (IHC):
Purpose: Used to analyze the localization and expression of FOXD proteins in CRC tissue sections.
Detailing: The manuscript should provide details on the IHC protocol, scoring criteria, and sample size.
Controls and Replicates:
The manuscript should discuss the use of appropriate controls and the number of biological and technical replicates for each experiment to ensure reliability and reproducibility.
Ethical Considerations:
A statement on ethical approval and informed consent for the use of human tissue samples should be included to ensure ethical compliance.
Statistical Analysis:
The manuscript employs appropriate statistical methods for data analysis, but it should provide more details on the specific tests used, significance thresholds, and how multiple testing was controlled.
Summary
The experimental design of the study is comprehensive and employs a combination of bioinformatic analyses and experimental validations. However, the manuscript would benefit from additional details on dataset selection, experimental protocols, controls, and ethical considerations. Providing these details would enhance the reproducibility, validity, and ethical integrity of the study.

Validity of the findings

Strengths in Validity:

Comprehensive Data Analysis:

The use of large-scale datasets from TCGA provides a robust foundation for the findings. TCGA is a well-regarded source of high-quality data for cancer research.
The integration of various bioinformatic tools (DESEQ2 for differential expression, Kaplan-Meier for survival analysis, CIBERSORT for immune infiltration) strengthens the overall analysis and supports the validity of the conclusions drawn.
Multiple Methods of Validation:
The study employs multiple experimental techniques (RT-qPCR, western blotting, immunohistochemistry) to validate the bioinformatic findings, which increases confidence in the results.
The use of different validation techniques helps to confirm that the observed gene expression changes are not artifacts of a single method.
Statistical Rigor:
The study uses appropriate statistical methods, such as ROC curves for diagnostic accuracy and Kaplan-Meier survival analysis, to assess the clinical significance of the FOXD subfamily genes.
The application of functional enrichment analysis provides further insight into the potential biological roles of these genes.
Weaknesses in Validity:
Lack of Detailed Methodological Information:
The manuscript does not provide detailed information on the selection criteria for datasets and samples, which could introduce bias and affect the reproducibility of the results.
Specific details on the parameters used for bioinformatic analyses (e.g., thresholds for differential expression, criteria for high and low expression groups in survival analysis) are missing, which are essential for replicating the study.
Control and Replicates:
The manuscript lacks detailed information on the controls and replicates used in experimental validation, such as the number of biological and technical replicates, and the use of appropriate controls. This information is crucial for assessing the reliability of the experimental findings.
Ethical Considerations:
There is no mention of ethical approval or informed consent for the use of human tissue samples. This is a significant oversight that can impact the ethical validity of the study.
Data Interpretation:
Some findings, such as the clinical significance of specific AUC values or the implications of immune infiltration results, are not thoroughly discussed. More detailed interpretation and contextualization within the existing literature are necessary to fully understand the impact of these results.
Potential Confounding Factors:
The study does not discuss potential confounding factors that could affect the findings, such as sample heterogeneity, variations in treatment regimens, or differences in patient demographics. Addressing these factors is important for ensuring the internal validity of the study.
Recommendations for Improving Validity
Methodological Transparency:
Provide detailed information on the selection criteria for datasets and samples, as well as the parameters used in bioinformatic analyses.
Include comprehensive descriptions of the experimental protocols, including the number of replicates and the use of controls.
Ethical Compliance:
Include a clear statement on ethical approval and informed consent for the use of human tissue samples to ensure ethical compliance.
Detailed Data Interpretation:
Offer a more thorough discussion of the findings, including potential clinical implications and comparisons with existing biomarkers for CRC.
Discuss the potential limitations of the study and the need for further validation in independent cohorts.
Address Confounding Factors:
Identify and discuss potential confounding factors that could impact the results and describe how these factors were controlled or accounted for in the analysis.
Summary
While the study demonstrates significant strengths in data analysis and validation, addressing the noted weaknesses is essential for ensuring the validity of the findings. By providing more methodological details, ensuring ethical compliance, and thoroughly interpreting the data, the study’s conclusions will be more robust and credible.

Additional comments

Nil

Reviewer 3 ·

Basic reporting

No comment.

Experimental design

No comment.

Validity of the findings

No comment.

Additional comments

In this article, the authors employed multiple bioinformatic methods to examine the relationship between the FOXD subfamily and CRC and obtained data from the TCGA. The manuscript is straightforward, well written, and concise. Definitely deserves to be published and is a valuable contribution to the “PeerJ” journal. However, the following comments need to be addressed in the introduction section, as recommended.

[1] “1.Introduction”, Line 45:
“The proportion of colorectal cancer(CRC) patients worldwide is rising [1].”.
From the epidemiological point of view, the authors should mention that very recently, new favorable subsets of cancers of undefined origin (CUP) seem to emerge including colorectal CUP. This new clinical entity is treated as CRC, and contributes to the current increased incidence of CRC.
Recommended reference: Rassy E, et al. New rising entities in cancer of unknown primary: Is there a real therapeutic benefit? Crit Rev Oncol Hematol. 2020 Mar;147:102882.

[2] “1.Introduction”, Lines 45-47:
“The causes of CRC are complex and diverse and may be associated with diet, family history, and genetic predispositions [2, 3].”.
The authors should make a comment about the population of elderly patients. Please, report that even though older patients are more prone to severe postoperative complications, there is no consensus that age affects survival outcomes. The prognosis of older patients may be confounded by differences in stage at presentation, tumor site, preexisting comorbidities, and type of treatment received.
Recommended reference: Osseis M, et al. Surgery for T4 Colorectal Cancer in Older Patients: Determinants of Outcomes. J Pers Med. 2022;12(9):1534.

[3] “1.Introduction”, Lines 47-49:
“Three major molecular pathways are involved in the pathogenesis of CRC: microsatellite instability acid phosphatase (MSI), chromosomal instability (CIN), and CpG island methylation phenotype (CIMP)[4].”.
At that point, the authors should mention that immune cell PD-L1 expression is significantly higher in MSI-H CRC as compared to MSI-L tumors, with no differences among the different MSI-H molecular subtypes. The recommended screening for defective, DNA mismatch repair includes immunohistochemistry (IHC) and/or MSI test. However, there are challenges in distilling the biological and technical heterogeneity of MSI testing down to usable data. It has been reported in the literature that IHC testing of the mismatch repair machinery may give different results for a given germline mutation and has been suggested that this may be due to somatic mutations.
Recommended reference: Adeleke S, et al. Microsatellite instability testing in colorectal patients with Lynch syndrome: lessons learned from a case report and how to avoid such pitfalls. Per Med. 2022;19(4):277-286.

[4] “1.Introduction”, Lines 50-52:
“Despite the development of modified treatments for CRC, including novel immunotherapy and targeted therapy, the desired overall survival rates have not been achieved [6].”.
At that point the authors should mention that FOLFOX-resistance in advanced CRC is significantly associated with upregulation and downregulation of several serum miRNAs. The differentiation of FOLFOX-resistant from FOLFOX responsive patients by serum miR-19a had a reported sensitivity and specificity of 66.7 and 63.9%, respectively. In terms of treatment response to anti-VEGF or anti-EGFR inhibitors in metastatic CRC, upregulation of miR-126 was correlated with bevacizumab resistance, whereas overexpression of miR-31, miR-100, miR-125b, and downregulation of miR-7, with resistance to cetuximab, respectively.
Recommended reference: Boussios S, et al. The Developing Story of Predictive Biomarkers in Colorectal Cancer. J Pers Med. 2019;9(1):12.

---

## Round 0.2 · Minor Revisions

Although the authors have made many of the revisions requested in the revision of the manuscript, there are still parts that need to be revised.

As per observations from one of the Section Editors:

"This subject appears to have been extensively studied, and there are some references not cited. The authors should explain how this study is different from:

Laissue P. The forkhead-box family of transcription factors: key molecular players in colorectal cancer pathogenesis. Mol Cancer. 2019 Jan 8;18(1):5. doi: 10.1186/s12943-019-0938-x. PMID: 30621735; PMCID: PMC6325735.

Hauptman, N., Jevšinek Skok, D., Spasovska, E. et al. Genes CEP55, FOXD3, FOXF2, GNAO1, GRIA4, and KCNA5 as potential diagnostic biomarkers in colorectal cancer. BMC Med Genomics 12, 54 (2019).
https://doi.org/10.1186/s12920-019-0501-z<https://doi.org/10.1186/s12920-019-0501-https://doi.org/10.1186/s12920-019-0501-z<https://doi.org/10.1186/s12920-019-0501-z

Kumegawa K, Yang L, Miyata K, Maruyama R. FOXD1 is associated with poor outcome and maintains tumor-promoting enhancer-gene programs in basal-like breast cancer. Front Oncol. 2023 May 10;13:1156111. doi:10.3389/fonc.2023.1156111. PMID: 37234983; PMCID: PMC10206236.

Lin, C.-H.; Lee, H.-H.; Chang, W.-M.; Lee, F.-P.; Chen, L.-C.; Lu, L.-S.; Lin, Y.-F. FOXD1 Repression Potentiates Radiation Effectiveness by Downregulating G3BP2 Expression and Promoting the Activation of TXNIP-Related Pathways in Oral Cancer. Cancers 2020, 12, 2690.
https://doi.org/10.3390/cancers12092690<https://doi.org/10.3390/cancers12092690><https://doi.org/10.3390/cancers12092690<https://doi.org/10.3390/cancers12092690>>;

Lin, C.-H.; Lee, H.-H.; Chang, W.-M.; Lee, F.-P.; Chen, L.-C.; Lu, L.-S.; Lin, Y.-F. FOXD1 Repression Potentiates Radiation Effectiveness by Downregulating G3BP2 Expression and Promoting the Activation of TXNIP-Related Pathways in Oral Cancer. Cancers 2020, 12, 2690.
https://doi.org/10.3390/cancers12092690<https://doi.org/10.3390/cancers12092690><https://doi.org/10.3390/cancers12092690<https://doi.org/10.3390/cancers12092690
https://journals.plos.org/plosone/article?id=10.1371/journal.pone.0127976<https://journals.plos.org/plosone/article?id=10.1371/journal.pone.0127976><https://journals.plos.org/plosone/article?id=10.1371/journal.pone.0127976 https://journals.plos.org/plosone/article?id=10.1371/journal.pone.0127976

Some of these references are not CLC, but since they are involved in other cancers, it is important to correlate them with the other cancers.

Also, on line 366, there is something wrong with the reference."

These suggested changes must be revised.

·

Basic reporting

The authors have made the suggested changes in the revised manuscript.

Experimental design

The authors have made the required corrections in the reviewed manuscript.

Validity of the findings

the authors have made the corrections as per reviewer's suggestion.

Reviewer 3 ·

Basic reporting

Adequate

Experimental design

Adequate

Validity of the findings

Adequate

---

## Round 0.3 · accepted · Accept

The authors have made the necessary corrections to their manuscripts in order for the article to be accepted for publication.